# Performance Comparison of Multiple Convolutional Neural Networks for Concrete Defects Classification

**DOI:** 10.3390/s22228714

**Published:** 2022-11-11

**Authors:** Palisa Arafin, Anas Issa, A. H. M. Muntasir Billah

**Affiliations:** 1Department of Civil Engineering, Lakehead University, Thunder Bay, ON P7B 5E1, Canada; 2Civil and Environmental Engineering Department, United Arab Emirates University, Al Ain P.O. Box 17551, Abu Dhabi, United Arab Emirates; 3Department of Civil Engineering, University of Calgary, Calgary, AB T2N 1N4, Canada

**Keywords:** structural health monitoring, concrete defects, deep learning, convolutional neural network, performance comparison, sensitivity analysis

## Abstract

Periodical vision-based inspection is a principal form of structural health monitoring (SHM) technique. Over the last decades, vision-based artificial intelligence (AI) has successfully facilitated an effortless inspection system owing to its exceptional ability of accuracy of defects’ pattern recognition. However, most deep learning (DL)-based methods detect one specific type of defect, whereas DL has a high proficiency in multiple object detection. This study developed a dataset of two types of defects, i.e., concrete crack and spalling, and applied various pre-built convolutional neural network (CNN) models, i.e., VGG-19, ResNet-50, InceptionV3, Xception, and MobileNetV2 to classify these concrete defects. The dataset developed for this study has one of the largest collections of original images of concrete crack and spalling and avoided the augmentation process to replicate a more real-world condition, which makes the dataset one of a kind. Moreover, a detailed sensitivity analysis of hyper-parameters (i.e., optimizers, learning rate) was conducted to compare the classification models’ performance and identify the optimal image classification condition for the best-performed CNN model. After analyzing all the models, InceptionV3 outperformed all the other models with an accuracy of 91%, precision of 83%, and recall of 100%. The InceptionV3 model performed best with optimizer stochastic gradient descent (SGD) and a learning rate of 0.001.

## 1. Introduction

Structural safety, reliability, and uninterrupted performance are vital concerns for maintaining the proper serviceability of any infrastructure. In today’s world, concrete is the most widely used construction material. However, structural health is highly disrupted due to the extreme environmental effect. Hence it is crucial to develop a systematical inspection system to maintain the structure’s serviceable condition. Furthermore, with the increasing number of aging infrastructures, frequent inspections are required to inspect the inherent damages and infer the potential causes of these damages, which provides essential guidance on structural assessment. Conventionally the site reconnaissance mostly depends on manual investigation, which is costly and labor-intensive. Moreover, this manual detection and identification of defects are time-consuming and subjective [1,2]. More importantly, the performance of the quantitative inspection of defects significantly depends on the inspector’s technical skill and experience. Therefore, an automatic concrete defects identification system with pre-built standards is highly recommended for efficient and objective defects assessment.

### 1.1. Prior Studies

Considering these shortcomings of the traditional visual inspection system, many researchers have introduced computer vision-based defects-identifying systems such as Image processing techniques (IPTs). Some of the most widely known approaches to IPTs for defects detection are the thresholding-based approach [3,4,5,6], and morphological methods [7,8]. The thresholding method converts an image into a binary image from color or grayscale image. Unfortunately, the performance of IPTs is negatively affected by lighting conditions and background obstructions of the images [9]. Therefore, to further improve the damage detection methods, some researchers have worked on combining Machine learning (ML) approaches with IPTs. Some of the previous studies included IPTs, the ML method (i.e., Support vector machine), and neural networks to categorize the damage features from other features [10,11]. Ref. [12] studied three unique principal component analysis (PCA) methods while working on automatic crack extraction in concrete structures.

Recently, the DL-based detection process has become popular for its automatic and optimized feature extraction potentiality [13]. Moreover, DL can analyze a large amount of data simultaneously with multiple categorizations, which facilitated damage evaluation in the SHM system [14]. In the context of SHM, DL techniques are implemented in the damage detection of infrastructure in three steps: classification, localization, and segmentation. DL has several techniques, such as deep belief networks (DBNs), recurrent neural networks (RNNs), encoder–decoder models, and convolutional neural networks (CNNs). CNNs have shown efficacy for defect detection purposes [15]. A CNN-based DL method was developed to detect concrete cracks by [16,17] and other defects by [18]. The CNN model has different algorithms based on the depth and width of the learning layers. Researchers continuously explore various CNN algorithms to develop an effective damage assessment model, such as VGGNet [19], ResNet [20], and Inception [21] models are used for damage pattern recognition and detection process. The authors of [22] compared three neural networks, AlexNet, VGGNet13, and ResNet18, to identify the concrete cracks and crack-free surfaces, with images of 227 × 227 pixels. The models were trained with 10,000 cracks and 10,000 crack-free images and obtained an accuracy of 97.6%, 98.3%, and 98.8%, respectively. This study used the data augmentation to create the 10,000 crack images database from 2000 original images. In the study of [23], AlexNet- and GoogLeNet-based models were used to identify the brick crack, spalling, and efflorescence. They used 1466, 1830, 865, and 984 original images of intact brick, crack, spalling, and efflorescence, respectively. Both models had an accuracy of over 90%. In another research, [24] compared five different types of base algorithms, VGGNet, ResNet, DenseNet, Inception, and MobileNet to identify the crack images of masonry surfaces. The dataset contained 351 images of cracks and 118 without any cracks and had a resolution of 224 × 224 pixels. The implementation of transfer learning brought a significant boost in the model’s performance, and with an accuracy of 95.3%, MobileNet outperformed all the other models. In a recent study of crack identification, [25] extensively analyzed the performance of transfer learning of pre-trained CNN models. This study also built a modified Dempster–Shafer (D–S) algorithm to improve the crack detection accuracy while proving robustness with images that contained various types of and intensities of noises.

### 1.2. Reserarch Objective and Contribution

Although DL approaches have proved to be exceedingly successful in image classification and automatic feature extraction, the in-depth review of prior studies has shown some existing limitations, only a few studies have worked with multiple damage detection. In contrast, multiple detections are essential to comprehend the actual scenario of damage condition of any structure. Even though some studies have worked with different types of damages [26,27], the image dataset is minimal for some cases, i.e., spalling, and rebar exposure. Moreover, most prior research lacks multiple CNN model analyses and detailed sensitivity analyses of hyper-parameters. A comprehensive comparison of the different CNN models’ performance based on a variety of hyper-parameters can provide a good understanding of how a well-tuned model can help build an automatic DL-based damage classification model. Considering the challenges above, this study outlined some specific improvements for all these challenges: (a) Building a large dataset of labelled images for two different types of defects representing the diversity of the defects’ physical parameters and image architectures. For CNN models, the successful completion of pattern recognition and object detection highly depends on a comprehensive and diverse dataset. Previously, most of researchers have developed and validated the CNN techniques with a limited quantity and defect-targeted images which do not replicate the real-time environmental exposure. In actual circumstances, it is highly unfavorable to collect images without any background noises because of the significant uncertainty in locations, lighting condition, and contents. This study collected defect images from various sources including actual industrial inspection reports (courtesy to TBT Engineering), web-based resources, and pre-developed datasets by other researchers. One of the primary focuses of this study is to build a comparative dataset collecting images from various resources to imitate the real structural site conditions. (b) Avoiding augmented images for the developed damage classification algorithm is another improvement. Augmented images are avoided as the augmented dataset can provide a false presentation of good performance with a specific dataset. At the same time, the models do not achieve a successful evaluation in real-world application. (c) Performing a detailed sensitivity analysis to identify optimized hyper-parameters for CNN classifiers and segmentation models is the final improvement. This study analyzes different pre-built CNN models for defect classification and segmentation. Moreover, the hyper-parameters are tuned during the training process to achieve an optimized CNN model. For sensitivity analysis, different types of hyper-parameters are selected and implemented in the CNN models to find the best-tuned hyper-parameters values for defects classification and detection.

## 2. Methodology

For defect classification, CNN models analyze the image pixel’s spectral information and classify the pixels into multiple classes. In this study, two different types of concrete defects are considered: (a) concrete crack and (b) concrete spalling. For the identification of these defects, the overall procedure is divided into three sections: (a) data processing, (b) CNN models training and tuning, and (c) trained model’s performance evaluation. A schematic diagram of the work flow followed in this study for defect identification is shown in Figure 1. From Figure 1, the CNN classification process is initialized with a data process which includes defects type selection, data acquisition from various resources, and image processing. Image processing indicates converting general image resolution into desired resolution and splitting the dataset for training and testing purposes. Once the data processing is completed, the defect images are used as input for CNN classifier models. In the CNN implementation stage, different parameters and hyper-parameters are trained and tuned to achieve the best performance from the models. Finally, at the evaluation stage the trained model is evaluated based on a few evaluation matrices and validated with a test dataset to check prediction accuracy.

### 2.1. Building Defects Database

A well-organized dataset containing quality and quantity is highly recommended for achieving a robust performance from any CNN model. According to [28], CNN models can achieve better test accuracy with a more extensive training dataset. While a larger dataset positively impacts the model’s performance, the dataset should have high-quality images representing real-world environmental conditions: Images with background noises, including surface roughness (i.e., scaling, edges, and holes), lighting condition, background debris, etc. The authors of [29] found that the quality and quantity of the dataset significantly influence the performance of the CNN model, and the low-quality images affect the models’ potentiality. In another study, ref. [16] tested a CNN model with a dataset of targeted and noise-free images. They subsequently tested it with a dataset of rough surface images and found that the model’s precision decreased from 87.4% to 23.1%.

#### 2.1.1. Data Preparation

In this study, concrete cracks and spalling are considered for defect types. One of the primary focuses of this study is to build a comparative dataset by collecting images from various resources to imitate the actual structural site conditions. Firstly, defect images were collected from actual infrastructure inspection reports executed by a local industry partner, TBT Engineering. These images served as an exact replication of an actual event that occurred at defected structure site. However, the number of images collected from the inspection reports is inadequate to run a successful DL-based automated defect condition assessment project. Therefore, this study took advantage of the online resources to deal with the challenges mentioned above, as some previous studies also explored DL applications in concrete defect identification. Part of the concrete crack and spalling images were retrieved from a freely available annotated dataset created by [30]. Apart from these sources, some images were collected from open-source online sources and experimental test results conducted on concrete members. Finally, a dataset of 4087 crack images and 1100 spalling images was developed for this study (Figure 2). A few data samples of crack and spalling images are presented in Figure 3. The developed dataset has a wide range of defects characteristics, such as different areas, lengths, widths, and shapes, including horizontal, vertical, and zigzag shapes on the various concrete surface. These realities in defect area and shape are meant to aid the CNN models in learning the versatile patterns of the defects to make a more accurate prediction with untrained images. Referencing Table 1, it can be stated that most studies have used the data augmentation process to create a big dataset from the original dataset. To the author’s best knowledge, the proposed dataset in this study is one of the most extensive datasets of both concrete defects without applying any image augmentation process.

#### 2.1.2. Data Processing

Since the images are collected from multiple sources, the image properties are different for the entire dataset. At first, the image resolutions are unified by converting all the images into a resolution of 224 × 224 pixels. Then, the entire dataset is randomly divided into input and testing images for a model’s learning process. The input dataset is used to train and develop a prediction model, whereas the function of the testing dataset is to determine the model’s prediction accuracy. The input images have two components: the training dataset and the validation dataset. While the training dataset was used for the learning process, the validation dataset offers an unbiased evaluation of the training dataset by subsequently tuning the hyper-parameters. Conventionally, while splitting the entire dataset, the input dataset is considered to have a more significant portion of images, while the rest was used for testing purposes. However, there is no universal approach to dataset splitting ratio. For instance, most of the researchers [2,32,33] have considered an 80–20% train–test split ratio for their CNN models. On the other hand, ref. [34] adopted 70% of the entire dataset as a training and validation dataset and the rest of the 30% as a test dataset. The authors of [35] divided the dataset into a 60–40% ratio to use 60% as input images and 40% for evaluating the models. As the crack and spalling datasets have a big difference in size, this study decided to use the maximum images for training and validation purposes for the CNNs classification and split the dataset into 70–20–10% ratios for training, validation, and testing purposes. Table 2 presents the summary of data distribution for train, validation, and testing. From the Table 2 it is evident that there is data imbalance between concrete cracks and spalling images. However, as crack and spalling have a very distinguishable features (i.e., defect area and shape) CNN models can easily identify the differences between the defects.

### 2.2. CNN Classifier Model Configuration

In the vision-based DL process, deep neural networks learn the features from the dataset by tuning a group of parameters and later on transferring these attributes to solve novel tasks. This phenomenon of transferring the learned data to a new model is referred to as transfer learning. In practical use, transfer learning uses the pre-learned elements from a trained model to initialize the training process of a new DL model. This can be considered as a less resource-intensive approach as the new models do not have to start training from scratch. To consider the pre-trained models for new tasks, usually the original model should have a certain amount of better generalization adaptability to perform satisfactorily with new unseen data [36]. In general, a novel CNN model requires analyzing a large amount of data resulting in training a few million parameters. However, these training parameters can reduce sizes by implementing a transfer learning process.

In this study, five different CNN classifiers are considered: (a) VGG19, (b) ResNet50, (c) InceptionV3, (d) Xception, and (e) MobileNetV2. One of the main reasons behind choosing these five models is that after analyzing the previous studies, it was perceived that these models have a consistent difference in their trainable layers and performance potentiality. For example, while the VGG-19, followed by ResNet-50, has the lowest trainable layers, they have shown acceptable performance prospects with their unique architecture. As MobileNetV2 was built to perform faster in a mobile application system, this model is considered in this study to evaluate the model’s damage identification performance if the model is implemented in a mobile application. Moreover, InceptionV3 and Xception have many trainable layers, which helped this study comprehend the variation in model performance with a change in trainable layers. The algorithms of these networks were developed using Keras applications [37]. Keras application includes the pre-built DL models, which can be used for training the model and making the prediction. For the coding language, Python is used backend by TensorFlow. After building the CNN classifier application, the model simulations are run using Google Collaboratory.

#### 2.2.1. VGG-19

In 2015, [19] proposed the VGG-16 and VGG-19 models and analyzed the effect of the depth of the CNN model for the classification purpose. VGG-19 consists of 19 layers with convolutional layers, pooling layers, fully connected layers, and a softmax layer. There are two distinctive characteristics of the VGG network: (a) the filter size remains the same for all the feature map sizes, and (b) using the max-pooling function the feature map size is reduced to half, and the number of filters obtained is doubled.

#### 2.2.2. ResNet-50

ResNet was first introduced by [20] where they described a residual learning algorithm with the advantage of going deeper without encountering performance degradation. ResNet was also proved effective in solving the problem with vanishing gradient descent by decreasing the error within the deeper layer. In each layer of the convolutional layer, a residual learning block was added, which worked as a “skip connection”.

#### 2.2.3. Inception

The Inception model was first introduced by [21] and showed remarkable performance on the ImageNet Visual Recognition Challenge (2014). This model was once regarded as the state-of-the-art deep learning model for its noteworthy performance in image recognition and detection. The main objective of this model is to connect the model sparsely, replacing the fully connected networks of the convolutional layers. The sparsely connected network is the core concept of the inception layer.

#### 2.2.4. Xception

The basic concept of Xception is based on the Inception and refers to “extreme inception”. However, Xception works in a reverse compared with Inception. Firstly, Xception applies the filters on each depth map, and a 1 × 1 convolution is used to compress the input space across the depth. Another notable difference between the Inception and Xception model is the presence of non-linearity. Inception uses non-linearity throughout all its operations, followed by ReLu non-linearity; however, Xception avoids any type of non-linearity in its architecture.

#### 2.2.5. MobileNetV2

This model takes a unique approach called depthwise separable convolutions to build a lightweight neural network. In practice, using the depthwise separable convolutions, MobileNet significantly reduces its quantity of the learnable parameters making the model smaller and faster. This unique convolution works in two steps: (a) depthwise convolution and (b) pointwise convolution. In depthwise convolution, the filters’ depth and spatial dimension (input channel) are separated, and a single filter is applied for each input channel. Finally, the pointwise convolution, a 1 × 1 convolution, combines the outputs of the depthwise convolution.

### 2.3. Sensitivity Analysis of Hyper-Parameters

In this study, pre-trained ImageNet weights are considered to start the training process of CNN models, followed by a continuous trial–error method to reach the optimized point of hyper-parameters. Then, a sensitivity analysis was performed to train the hyper-parameters and find the best-performed models. This study considers a few hyper-parameters, such as batch size, activation function, optimization function, loss function and learning rates for sensitivity analysis. As [38] mentioned, these parameters are the most critical parameters that guide the models toward optimized convergence. The details of these hyper-parameters are presented in Table 3.

The feature extraction in DL is a nonlinear process and requires the application of nonlinear functions called the activation function. In a neural network, the activation layer uses an activation function (nonlinear) to navigate how the weighted sum of the input transforms from nodes to output. In this study, Rectified Linear Activation (ReLu) function (Figure 4) is used for all the CNN models, as shown in Equation (1). ReLu is a linear function that provides output only if the input is positive; otherwise, the output is zero meaning the neuron is deactivated. This provides advantages to computational efficiency as not all the neurons are activated in one instance.
f(x) = max (0, x) (1)

To update the model variables, it is crucial to calculate the derivation of the ground truth and the prediction value. The function that calculates the derivation is referred to as the loss function. This study considers the binary cross-entropy (BCE) loss function for the classification. BCE is a cross-entropy function used to choose between two choices (i.e., concrete crack and spalling). This loss function is usually considered to achieve prediction by the sigmoid activation function. Cross-entropy (CE) is a pixel-wise loss function and performed prominently for various object detection applications [39]. Moreover, using this loss function in the CNN model provides the model the highest compatibility to be employed in the new dataset. Equation (2) presents the mathematical approach to how the binary cross-entropy loss function (LBCE) calculates the average loss, where y_j_ is the scaler value of output, y_i_ is the corresponding target value, and n is the output size.
(2)$$\mathrm{Loss}=-\frac{1}{n}\sum_{i=1}^{n}y_{i}\times \log{\;y}_{j}+\left(1-{\;y}_{i}\right)\times \log\left(1-{\;y}_{j}\right)$$

The optimization technique in a neural network works by finding the minimum or maximum output of the function depending on the input parameters or arguments. While updating the variable parameters through the forward pass and backpropagation process, the model emphasizes minimizing the loss function and optimizing the model’s accuracy. The loss function guides the optimizers by quantifying the difference between the expected result and the predicted result of the model. For classification CNN, two optimizers are used: Stochastic Gradient Descent (SGD) and Root Mean Square Propagation (RMSprop). SGD is a type of gradient descent process that is linked with a random probability. SGD takes a single random data to update its parameters for each iteration. To the DL researchers, RMSprop is one of the most popular optimizers. RMSprop has a unique feature which restrains swaying in the vertical direction when helping the learning rate to learn faster in the horizontal direction, making the convergence faster.

To achieve the best output result, the values of hyper-parameters for CNN classifiers are designated after carefully analyzing the learning process. According to Table 2, a batch size of 10 is considered, and the models are trained for 100 epochs. An epoch refers to the one complete training cycle of a forward pass and backpropagation. To finalize the epoch size, two functions called early stopping and the reduced learning rate is applied in these models. These two functions help the models avoid over-fitting by stopping the model’s training process when the best accuracy is achieved. This also helps reduce the models’ computational costs (time and computer memory). After completing all the combinations of sensitivity analysis, it is found that the models reach their optimized performance condition within the 100 epochs. Therefore, this study considered 100 epochs for model training. Moreover, for batch size, it is observed that with a group of 10 images, the model learns the features with a minimal computational cost. Moreover, as an activation function, ReLu performed to have a positive impact on the model’s performance. According to some previous studies, SGD and RMSprop are some commonly used optimizers to train the CNN models [40,41,42,43]. Moreover, some studies used a learning rate of 0.001 [40] and 0.0001 [42] to control the learning process of the CNN model to achieve the best performance. Hence this study explored two different optimization functions: SGD and RMSprop along with three different learning rates 0.1, 0.001, and 0.0001 for each of the five models separately and summarized the results in Section 3. Finally, the best hyper-parameters values are decided on by evaluating the trained model with the testing dataset and comparing their results using the evaluation matrices.

### 2.4. Evaluation Metrics

In CNN model analysis, evaluation matrices are considered to quantify the statistical performance of the output results of the trained models. Evaluating the DL models is essential to understanding the output results and comparing various models’ performance to select an appropriate model for different tasks. This study considers four different metrics to evaluate the performance of defects classification: Accuracy, Precision, Recall, and Confusion matrix. The following are formulations for these evaluation metrics:(3)$$\mathrm{Accuracy}=\frac{\mathrm{TP}+\mathrm{TN}}{\mathrm{TP}+\mathrm{TN}+\mathrm{FP}+\mathrm{FN}}$$
(4)$$\mathrm{Precision}=\frac{\mathrm{TP}}{\mathrm{TP}+\mathrm{FP}}$$
(5)$$\mathrm{Recall}=\frac{\mathrm{TP}}{\mathrm{TP}+\mathrm{FN}}\;$$

Here, TP, TN, FP, and FN indicate true positive, true negative, false positive, and false negative, respectively. TP denotes if the crack image is classified correctly while TN shows if the spalling image is classified correctly. FP denotes if the crack image is classified incorrectly while FN represents if the spalling image is classified incorrectly.

The confusion matrix is a type of matrix which presents the numerical summary of the final predictions (TP, TN, FP, and FN). This model uses a binary confusion matrix, dividing the dataset into two classes. For this study, “0” represents the “crack” and “1” is termed as “spalling”.

## 3. Result and Discussion

For the sensitivity analysis, at first, each CNN classifier model, VGG19, ResNet50, IncptionV3, Xception, and MobileNetV2, considered two different optimizers, SGD and RMSprop. Later, each CNN classifier with both SGD and RMSprop optimizer was evaluated for three learning rates 0.0001, 0.001, and 0.1. Finally, thirty models were analyzed and evaluated separately with the combination of two optimizers and three learning rates for five CNN classifiers. Table 4 represents the performance of all the CNN classifiers for learning rates 0.0001, 0.001, and 0.1, respectively. Three evaluation matrices, accuracy, precision, and recall, were considered to evaluate the model’s performance.

From Table 4, it can be established that InceptionV3 outperformed all the other models in the case of both optimizers. For a learning rate of 0.001, SGD optimizer IncpetionV3 achieved the best accuracy, precision, and recall values of 91%, 82%, and 100%, respectively. Xception attained the second-best performance by adopting SGD optimizer with an accuracy of 89%, precision of 82%, and recall of 94%. The architecture of Xception is based on the Inception model, which is one of the possible reasons for the performance resemblance of these two models. The inception model is considered to have better performance than ResNet as the Inception model focuses on reducing computation cost while learning the features with deeper learnable layers, eventually increasing optimization accuracy. However, ResNet only works on computational accuracy without concern for optimization, which can overfit the training process and ultimately affect the prediction performance. In the case of MobileNet, this model has fewer learnable parameters than the Inception model, which can be an advantage to achieving good performance with lower memory capacity, but then with higher learnable parameters the Inception model performs better than MobileNet. It is evident that the InceptionV3 model outranked the other models.

In the case of a learning rate of 0.001, InceptionV3 showed the best output, followed by the Xception model (Figure 5). With a learning rate of 0.1, the training process skipped many learning features and converged faster towards a suboptimal position. In contrast, a learning rate of 0.0001 is a slow pace to update the models’ weights and increase the computational cost without improving the model’s performance significantly. Between two optimizers, the optimizer SGD aided in obtaining the best performance for defects classification for IncpetionV3. The accuracy, precision, and recall values of InceptionV3 are found at 91%, 83%, and 100%, respectively. Similar to InceptionV3, Xception has the best performance with the SGD optimizer. According to [44] SGD has better stability and generalization capacity than other adaptive optimization methods (i.e., RMSprop), which helps the models to reach their optimization point better than others. The authors of [45] studied experimental and empirical analysis to prove that for classification tasks, SGD converged better than other adaptive methods. They also stated that the performance did not improve much with faster initial training progress in validation.

After analyzing the models with evaluation metrics, confusion matrix evaluation was performed based on the true label vs. prediction label of crack and spalling to determine which CNN classifier performs better on defects identification. Figure 6 portrays the confusion matrix for InceptionV3 and Xception models. For all the confusion matrix diagrams, the x-axis and y-axis represented the true label and predicted label, where “0” denotes the crack, and “1” refers to “spalling”. As mentioned earlier, the InceptonV3 and Xception model has the best performance with optimizer SGD and a learning rate of 0.001. Therefore, this study illustrated the confusion matrix graphs only for those conditions. Figure 6a,b illustrate the true and false prediction of defects by the InceptionV3 and Xception model, respectively, for a learning rate of 0.001 and optimizer SGD. The graphs show that crack prediction with both InceptionV3 and Xception models predicted forty-nine images correctly while making eight false predictions. In the case of spalling detection, InceptionV3 predicted all the spalling cases correctly, whereas Xception falsely identified two. From the explanation above, it is clear that the InceptionV3 model has superiority over the Xception model.

As the loss function helps the model reduce the difference between the true value and prediction value for tuning the hyper-parameters, it is essential to track the training loss and validation loss over the training period. Figure 7, Figure 8 and Figure 9 present the graphical understanding of the InceptionV3 model’s performance over the epochs for three learning rates 0.0001, 0.001, and 0.1. From the graphs, it is prominent that IncpetionV3 models have the least amount of loss with a learning rate of 0.001. Moreover, the trained model obtained sharp training accuracy, precision, and recall close to 100%.

After analyzing the model’s performances, the IceptionV3 model ranked as the best-performed model for defects classification. Moreover, this model reached its performance-optimized point owing to the SGD optimization function and learning rate of 0.001. Apart from InceptionV3, the Xception model also showed a promising ground for defect classification using the SGD optimizer. On the other hand, among all the CNN classifiers VGG19 ranked last. One possible reason behind the InceptionV3 model functioning better than other models is that the model has the highest layers of depth for learning, which facilitates the model to gain better performance. On the other hand, VGG19 has the least depth of learning layers, which may have affected its overall performance.

Figure 10 and Figure 11 demonstrate some sample results of defects identification of cracks and spalling for all the CNN classifiers. The image’s first sentence describes the prediction result of the defects, and the second line shows the label of defects type. Figure 10 indicates that InceptionV3 predicted most of the cracks with 100% accuracy. On the other hand, some crack images have an accuracy of around 90% and predicted very few crack images with spalling. Moreover, the VGG19 model has the least accuracy in crack prediction and even has some false predictions. Figure 11 shows that similar to crack prediction, the InceptionV3 model also performed best for spalling detection and VGG19 has the least accuracy. In both figures, the red box indicates the prediction inaccuracy. All the probability percentages for each damage case are the output results of developed CNN models.

## 4. Conclusions

This research investigated the performance of various DL methods for automatic damage detection on concrete surfaces. For defects classification, this study conducted CNN classification for multi-class defects; concrete crack and spalling trained the model with different types and values of hyper-parameters to obtain the best output from the CNN classifiers. This study collected a dataset of 4080 crack images and 1100 spalling images, which is one of the largest datasets of both concrete defects without applying any image augmentation process. The conclusions drawn from this study are summarized below:A total of thirty models were evaluated combining the learning rates (0.0001, 0.001, and 0.1) and optimization functions (SGD and RMSprop) with five different CNN models (VGG-19, ResNet50, MobileNetV2, Xception, and InceptionV3);InceptionV3 model outranked the other models with accuracy, precision, and recall of 91%, 83%, and 100%, respectively. One possible reason behind the InceptionV3 model functioning better than other models is that the model has the highest layers of depth for learning, which facilitates the model to gain better performance. VGG19 has the least prospect with defect identification;With the help of the confusion matrix, this study found that IncpetionV3 made the least false predictions with crack identification. Moreover, IncpetionV3 labelled all the spalling cases correctly in the case of spalling identification;Among three learning rates, 0.0001, 0.001, and 0.1, with a learning rate of 0.001 all the CNN models achieved the best performance, which establishes the idea that a low learning rate does not always confirm better performance with CNN models;In the case of optimization functions, SGD assisted the CNN modes to achieve better performance, proving that SGD has better stability and generalization capacity than other adaptive optimization methods (i.e., RMSprop).

## 5. Recommendation for Future Studies

Based on the analysis performed in this study, a few areas have future scopes to improve the automatic defects detection process. Firstly, this study worked with images of two types of defects because of the limited availability of resources for other types of defects. Future projects working with DL-based defects detection need more collaboration with industrial partners to collect a large amount of diverse images. Secondly, future studies can take advantage of DL’s multiple object detection proficiency and create a model capable to identify multiple defects at a time from both images and videos. Once an adequate dataset is developed, it is possible to identify various types of defects from a single image or a video clip. Moreover, the outcomes of this research work are expected to expedite future research toward optimizing the CNN models to develop an automatic damage detection process with real-world application.

## Figures and Tables

**Figure 1 sensors-22-08714-f001:**
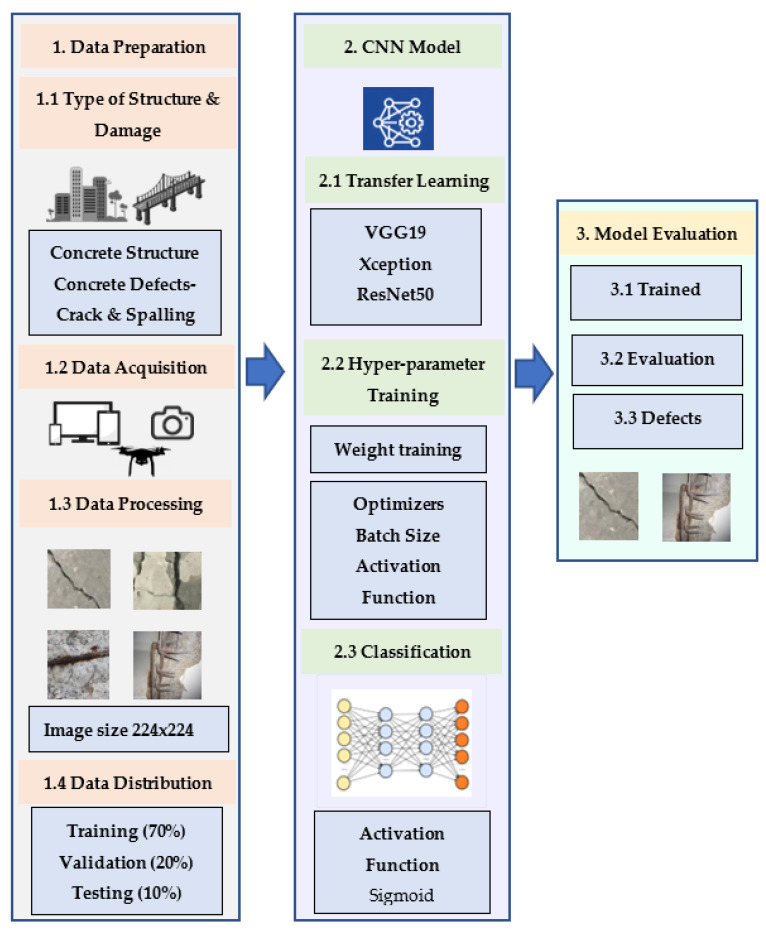
Workflow chart for defects classification.

**Figure 2 sensors-22-08714-f002:**
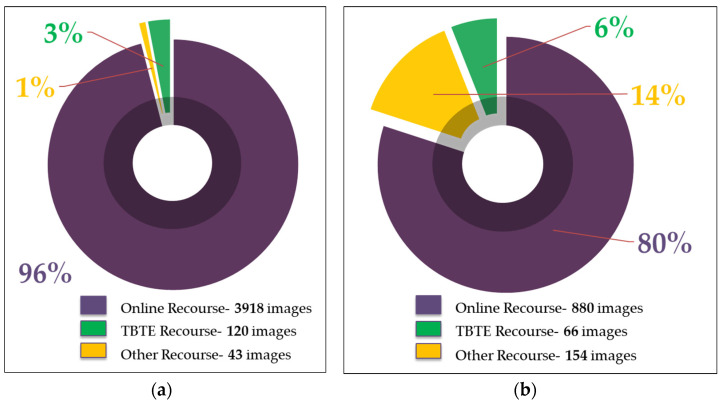
Size of dataset. (**a**) Crack dataset. (**b**) Spalling dataset.

**Figure 3 sensors-22-08714-f003:**
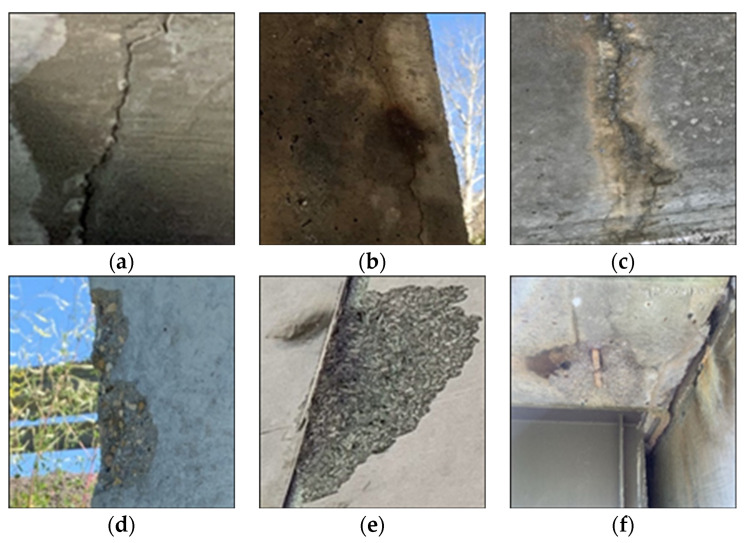
Sample images for defects dataset (**a**–**c**) crack and (**d**–**f**) spalling (image courtesy: TBT Engineering).

**Figure 4 sensors-22-08714-f004:**
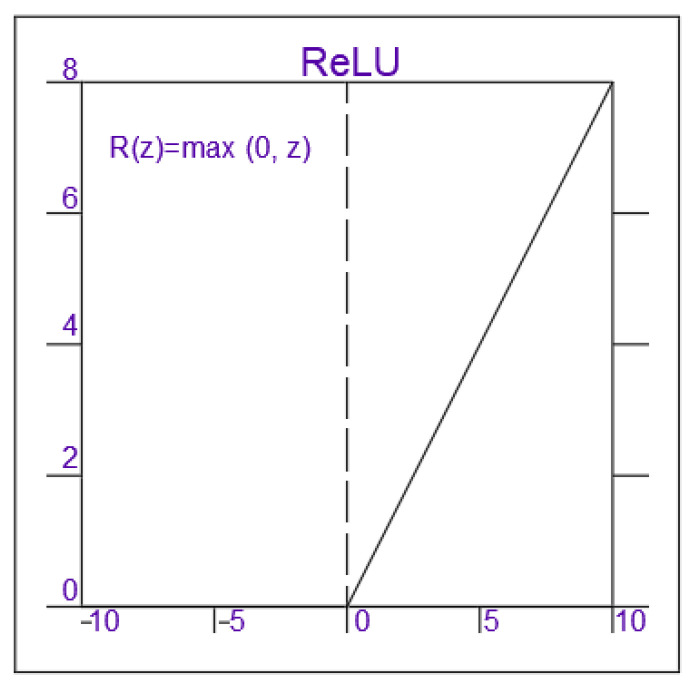
Activation function (ReLu).

**Figure 5 sensors-22-08714-f005:**
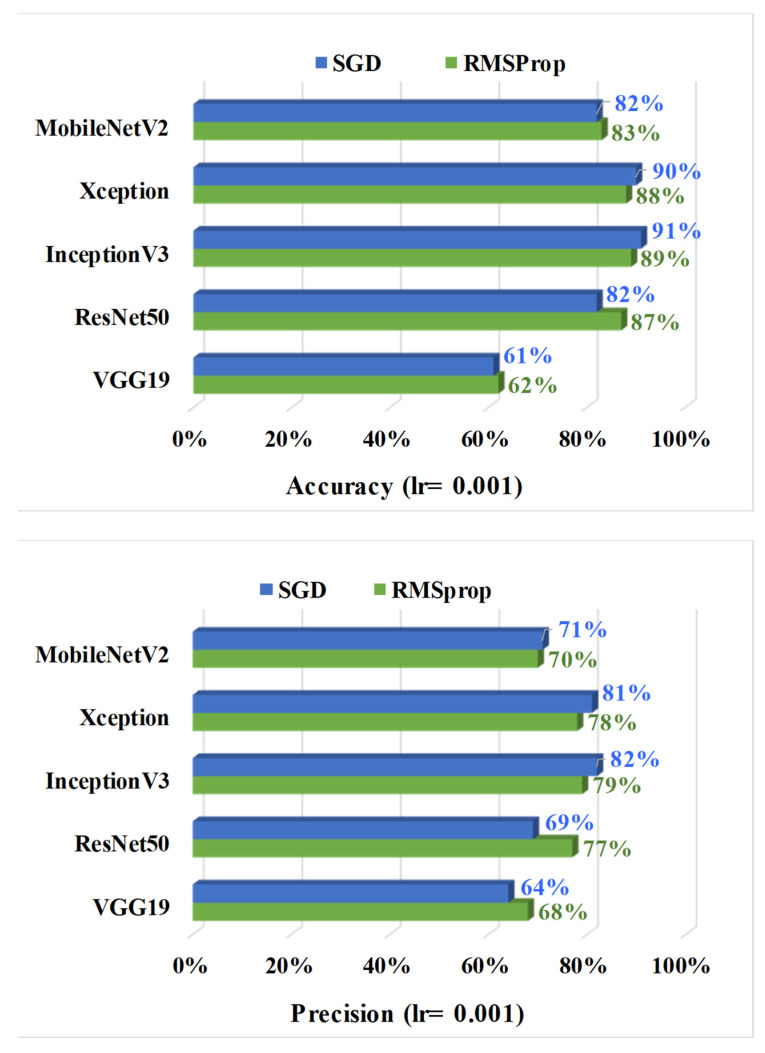
Defects classification: comparison of the evaluation metrics based on SGD and RMSprop optimization function for learning rate 0.001.

**Figure 6 sensors-22-08714-f006:**
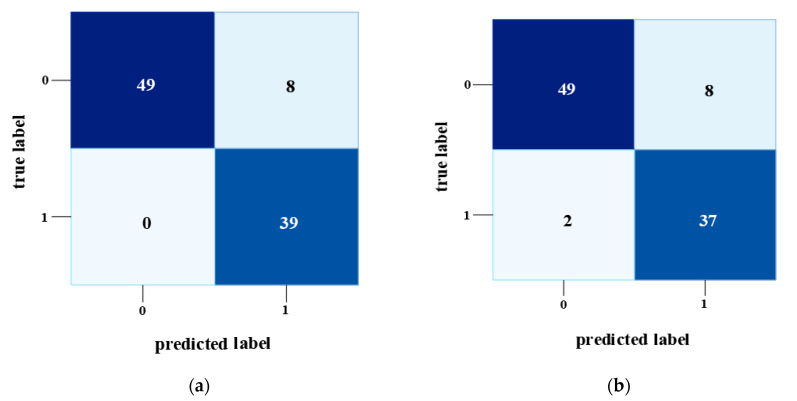
Confusion matrix for InceptionV3 and Xception for optimizer SGD and learning rate 0.001. (**a**) InceptionV3 (SGD). (**b**) Xception (SGD).

**Figure 7 sensors-22-08714-f007:**
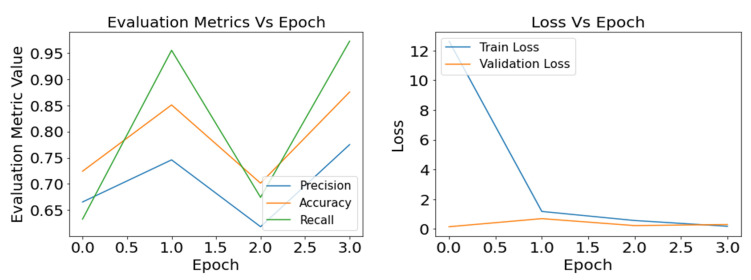
Defects classification with InceptionV3: result of the evaluation metrics and model loss for optimizer SGD and learning rate 0.0001.

**Figure 8 sensors-22-08714-f008:**
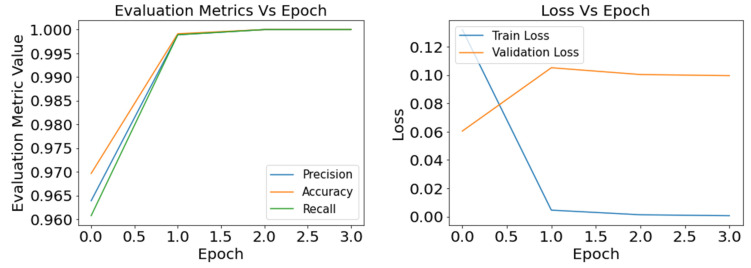
Defects classification with InceptionV3: result of the evaluation metrics and model loss for optimizer SGD and learning rate 0.001.

**Figure 9 sensors-22-08714-f009:**
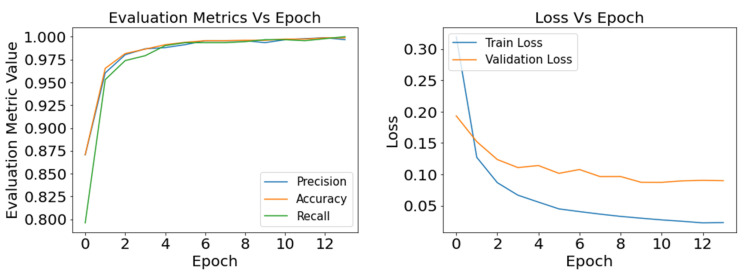
Defects classification with InceptionV3: result of the evaluation metrics and model loss for optimizer SGD and learning rate 0.01.

**Figure 10 sensors-22-08714-f010:**
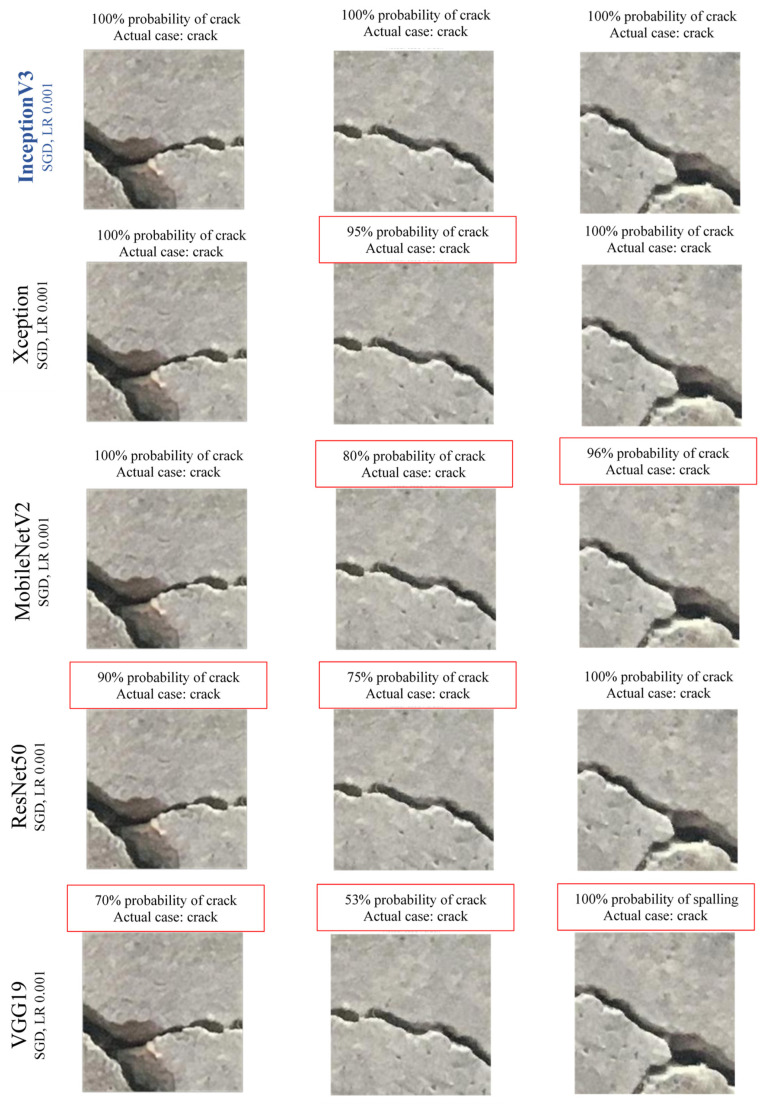
Sample images for crack prediction using CNN models.

**Figure 11 sensors-22-08714-f011:**
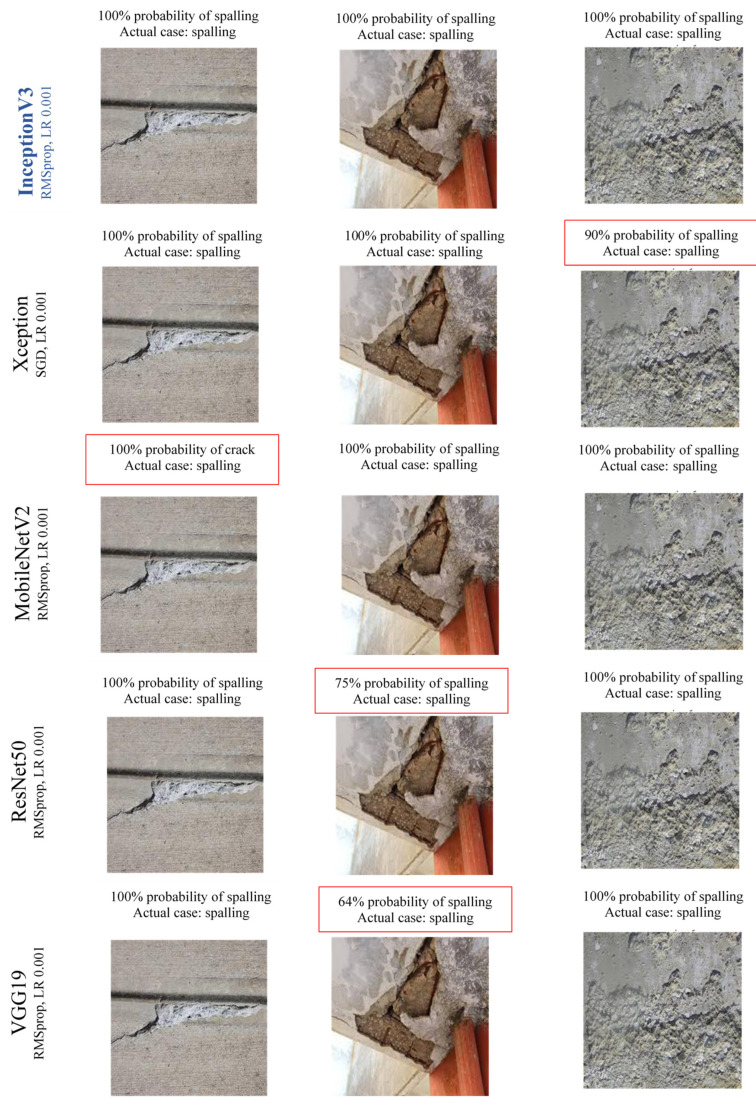
Sample images for spalling prediction using CNN.

**Table 1 sensors-22-08714-t001:** Dataset comparison of prior studies and current study.

Reference	Original Dataset Size	Defects Type	Data Augmentation
[13]	808 cracks, 86 non-cracks	Crack or non-crack	No
[22]	2000 cracks (road and bridge)	Crack or non-crack	Yes
[24]	351 cracks 118 non-cracks (masonry)	Crack or non-crack	Yes
[2]	1800 cracks	Crack or non-crack	No
[31]	1184 cracks (pavement)	Different types of cracks	Yes
Proposed dataset	4087 cracks, 1100 spalling	Crack and spalling	No

**Table 2 sensors-22-08714-t002:** Image distribution for classification CNNs.

Defect Classes	Total	Train Dataset	Validation Dataset	Test Dataset
Crack	4087	2861 (70%)	817 (20%)	409 (10%)
Spalling	1100	770 (70%)	220 (20%)	110 (10%)

**Table 3 sensors-22-08714-t003:** Details of hyper-parameters.

Name of Parameters	Value of Parameters
Batch Size (CNN classifiers)	10
Learning rate (CNN classifiers)	0.1, 0.001, 0.0001
Optimization function	SGD, RMSprop
Activation function	ReLu
Evaluation metrics threshold	0.5
Loss function (CNN classifiers)	Binary cross-entropy
Pre-trained weights	ImageNet
Callbacks	Early stopping
Epoch	100

**Table 4 sensors-22-08714-t004:** Summary results of defects classification models.

CNN Models	Learning Rate	Accuracy		Precision		Recall	
		SGD	RMSProp	SGD	RMSProp	SGD	RMSProp
*** InceptionV3**	0.1	86%	88%	78%	82%	100%	97%
	**0.001**	**91%**	**89%**	**83%**	**79%**	**100%**	**100%**
	0.0001	84%	89%	81%	84%	94%	100%
Xception	0.1	89%	87%	79%	76%	100%	100%
	0.001	90%	88%	81%	78%	100%	100%
	0.0001	89%	88%	82%	78%	94%	100%
MobileNetV2	0.1	81%	79%	71%	73%	94%	76%
	0.001	82%	83%	71%	72%	94%	100%
	0.0001	82%	84%	71%	70%	94%	100%
ResNet-50	0.1	85%	87%	72%	76%	97%	100%
	0.001	82%	87%	69%	77%	89%	97%
	0.0001	79%	85%	69%	74%	89%	97%
VGG-19	0.1	63%	65%	60%	62%	81%	82%
	0.001	61%	62%	64%	68%	80%	84%
	0.0001	63%	67%	61%	62%	82%	86%

Note: * Best performance.

## Data Availability

The data presented in this study are available on request from the corresponding author. The data are not publicly available due to confidentiality agreement with industry partner.

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
