# Peer review of "Performance Comparison of Multiple Convolutional Neural Networks for Concrete Defects Classification"

_sensors, 2022, doi:10.3390/s22228714_

Round 1

Reviewer 1 Report

In this paper, some existing convolutional (CNN) models such as VGG-19, ResNet-50, Inception V3, Xception and MobileNetV2 have been used to classify two types of concrete defects. Some technical contributions can be observed in terms of the constructions of new datasets for these two types of concert defects. Furthermore, the authors have also demonstrated the network performances under different hyperparameter settings. The manuscript is well written in overall and easy to understand. Some meaningful conclusions have also been drawn based on the simulation results. However, there are some major comments need to be addressed by the authors summarized as follows:

1. The title of this paper does not really reflect the nature of works. There are only two types of concrete defects are considered, hence it is not appropriate to claim as "multiple". The keywords of "Performance Comparisons" should be mentioned in the manuscript title since there no new CNN model developed in this paper.

2. Job designations of authors are not required to be included in affiliation. 

3. In Lines 162 and 163, the authors have claimed that their dataset of both concretes defects is one of the most extensive datasets built. To validate this statement, a table should be used to compare and summarize the datasets used by existing works, i.e., [13]-[24].

4. Authors are suggested to share the datasets so that other researchers can be benefited.

5.  Typo is found in Figure 1, i.e., "Imag". Some alphabets are hide by the boxes. Please rectify these issues.

6. From Table 1, there are 4087 and 1100 images with crack and spalling defects, respectively. Since the number of crack images are almost 4 times more than spalling images, authors need to justify if this will cause data imbalance issue. If yes, what will be the strategy to handle it?

7. Some brief descriptions should be provided to the CNN models used in current study. 

8. ROC curves and AUC scores provided by CNN models should be compared in current study. Furthermore, F1 score produced by different CNN models should be compared since the datasets used might have data imbalanced issue. 

9. Lines 342-345: The definitions of TP, TN, FP and FN can be included in the sentence instead of listed individually. 

10. Quality of figures 5 to 9 are not good enough. Please improve their resolution and size for better readability. 

11. Why only compare the confusion matrices produced by Inception V3 and Xception? It is more appropriate to present the confusion matrices of other CNN models.

12. Sentence appears in Line 549 seems unnecessary.  

Reviewer 2 Report

This manuscript investigated the deep learning models for multiple concrete crack classification, where VGG-19, ResNet-50, InceptionV3, Xception and MobileNetV2 were developed for the task of interest. Various concrete surface image datasets were combined to validate the performance of the proposed models. Overall, the topic of this research is interesting, and the manuscript was well organised and written. The detailed comments are summarised as follows.

1.       The contribution and innovation of the manuscript should be clarified clearly in abstract and introduction.

2.       Broaden and update literature review on application of CNN or deep learning in civil materials/concrete area, such as crack detection. E.g. Torsional capacity evaluation of RC beams using an improved bird swarm algorithm optimised 2D convolutional neural network; Vision-based concrete crack detection using a hybrid framework considering noise effect.

3.       The performance of deep learning models is heavily dependent on the setting of hyperparameters. How did the authors set them in this research to achieve optimal classification accuracy?

4.       In this study, the authors used several pre-trained deep CNN models for concrete crack classification application. Accordingly, transfer learning technology was used to transform the pre-trained networks to models that can be used for crack classification. Accordingly, more information about transfer learning should be provided.

5.       Fig. 6: the confusion matrices of all the networks should be presented.

6.       How about the robustness of the proposed method against noise effect?

7.       More future research should be included in conclusion part. 

Reviewer 3 Report

In this article, the authors propose an approach based on Convolutional Neural Networks for the  Classification of Concrete Defects.   Comments, Remarks, and Suggestions: 1. The article is well-written and well-structured. 2. The obtained results are good. 3. It would be better to devote a separate section to related works. 4. The authors may include a table that summarizes related works and makes a comparison with them. 5. The authors need to add a short paragraph in the introduction which summarizes their main contributions. 6. Similarly, they need to add another paragraph that describes the structure of the paper. 7. The paper contains too many Figures. Some of them may be placed in the appendix. 8. The conclusion is a bit long and may be separated into two sections: Discussion and Conclusion. 9. It looks that line 549 was inserted by mistake. 10. In future work, the authors may consider photos and videos taken by smartphones by different users in order to build larger datasets. For this purpose, the authors may consider the following article and include it in references: - https://ieeexplore.ieee.org/document/9327468

Round 2

Reviewer 1 Report

The authors have addressed majority of comments provided. The justifications provided by authors for not addressing comment #8 is acceptable and convincing. I recommend this paper to be accepted.

Reviewer 2 Report

All the technical issues have been well addressed by the authors.